# Association of Skin Microbiome with the Onset and Recurrence of Pressure Injury in Bedridden Elderly People

**DOI:** 10.3390/microorganisms9081603

**Published:** 2021-07-27

**Authors:** Shigefumi Okamoto, Kazuhiro Ogai, Kanae Mukai, Junko Sugama

**Affiliations:** 1Advanced Health Care Science Research Unit, Innovative Integrated Bio-Research Core, Institute for Frontier Science Initiative, Kanazawa University, Kanazawa 920-0942, Japan; junko.sugama@fujita-hu.ac.jp; 2Department of Clinical Laboratory Sciences, Faculty of Health Sciences, Institute of Medical, Pharmaceutical, and Health Sciences, Kanazawa University, Kanazawa 920-0942, Japan; 3AI Hospital/Macro Signal Dynamics Research and Development Center, Institute of Medical, Pharmaceutical, and Health Sciences, Kanazawa University, Kanazawa 920-0942, Japan; kazuhiro@staff.kanazawa-u.ac.jp; 4Department of Clinical Nursing, Faculty of Health Sciences, Institute of Medical, Pharmaceutical and Health Sciences, Kanazawa University, Kanazawa 920-0942, Japan; kanae_m@staff.kanazawa-u.ac.jp; 5Research Center for Implementation Nursing Science Initiative, Fujita Health University, Toyoake 470-1192, Japan

**Keywords:** pressure injury, bedridden elderly, skin microbiome, infection, *Staphylococcus*

## Abstract

Pressure injuries have been identified as one of the main health hazards among bedridden elderly people. Bedridden elderly people often stay in the same position for a long time, because they cannot switch positions; thus, the blood flow in the part of the body that is being compressed between the bed and their own weight is continuously blocked. As a result, redness and ulcers occur due to lacking oxygen and nutrients in the skin tissues, and these sites are often infected with microorganisms and, thus, become suppurative wounds, a condition commonly determined as pressure injuries. If left untreated, the pressure injury will recur with microbial infections, often resulting in cellulitis, osteomyelitis, and sepsis. The skin microbiome, in which many types of bacteria coexist, is formed on the skin surface. However, it remains unclear what characteristic of the skin microbiome among the bedridden elderly constitutes the development and severity of pressure injuries and the development of post-pressure injury infections. Thus, in this review article, we outlined the changes in the skin microbiome among the bedridden elderly people and their potential involvement in the onset and recurrence of pressure injuries.

## 1. Introduction

### 1.1. Pressure Injuries: A Serious Health Problem among Bedridden Elderly People

According to the 2019 White Paper on Aging Society released by the Japanese government, the aging rate in Japan is currently at 28.1%, and the population aged 75 and over has been estimated to be 17.98 million, accounting for 14.2% of the country’s total population [1]. These proportions are higher compared to other countries, making Japan the number one super-aging society in the world. Most elderly people cannot live on their own for several years before their death, thus needing support and long-term care. According to the Longevity Science Promotion Foundation in Japan, the period when support and care by a supporter is necessary is about 9 years for men and about 12 years for women, and various cares for the elderly during that period have become a major problem. In particular, the number of elderly people who are forced to live in a bedridden state is increasing year by year, and it has become a significant problem in how to prevent a worsening quality of life and health disorders among this specific population.

People who are in a bedridden state often have many health problems. A typical example is pressure injuries. Bedridden elderly people during their sleep often cannot change their positions [2,3,4,5,6,7,8,9]. Therefore, bedridden elderly people often stay in the same position for a long time during their sleep, thus making the blood flow in the part of their body that is compressed by their own weight from the bed continuously blocked [9,10]. Then, oxygen and nutrients in the continuously compressed skin tissue become low, resulting in redness, ulcerations, and suppurative wounds by pathogenic microorganism infections in the skin fragile site [2,3,4,5,6,7,8,9,10,11], a skin condition referred to as pressure injuries.

The most common site of a pressure injury is the area that is in direct contact with the bed and is continuously compressed, i.e., back of head, ear, shoulder, elbow, hip area, sacrum, heel, and ankle [12,13]. External factors (pressure acting perpendicular to the tissue and stress against tension and shear caused by friction between the tissue and the supporting surface) and internal factors (aging, undernutrition, paralysis, various skin abnormalities, etc.) have been determined to be the causative factors of pressure ulcers [12,13]. Especially in the case of undernutrition, the risk of developing a pressure injury remains high [12,13]. In addition, the symptoms of pressure injuries only become severe if left untreated.

One indicator of pressure injury progression is the National Pressure Ulcer Advisory Panel (NPUAP) classification [14], which categorizes pressure injuries into the following: suspected Stage 1 pressure injury, Stage 2 pressure injury, Stage 3 pressure injury, Stage 4 pressure injury, Unstageable pressure injury, and Deep tissue pressure injury. Stage 1 pressure injury indicates intact skin with a localized area of non-blanchable erythema, which may appear differently in darkly pigmented skin. The presence of blanchable erythema or changes in sensation, temperature, or firmness may precede visual changes. Color changes do not include purple or maroon discoloration. Stage 2 pressure injury indicates a partial-thickness loss of skin with exposed dermis. The wound bed is viable, pink or red, moist, and may also present as an intact or ruptured serum-filled blister. Adipose is not visible, and deeper tissue is not visible. Granulation tissue, slough and eschar, are not present. These injuries commonly result from adverse microclimates and shear in the skin over the pelvis and shear in the heel. Stage 3 pressure injuries indicate full-thickness skin losses, in which adipose is visible in the ulcer and granulation tissue, and epibole (rolled wound edges) is often present. Slough and/or eschar may be visible. The depth of the tissue damage varies by anatomical location; areas of significant adiposity can develop deep wounds. Undermining and tunneling may occur. Fascia, muscle, tendon, ligament, cartilage, or bone is not exposed. If slough or eschar obscures the extent of tissue loss, this is an unstageable pressure injury. Stage 4 pressure injuries indicate that full-thickness skin and tissue loss with exposed or directly palpable fascia, muscle, tendon, ligament, cartilage, or bone in the ulcer. Slough and/or eschar may be visible. Epibole (rolled edges), undermining, and/or tunneling often occur. The depths vary by anatomical location. If slough or eschar obscures the extent of the tissue loss, this is an unstageable pressure injury. Unstageable pressure injuries indicate full-thickness skin and tissue loss, in which the extent of tissue damage within the ulcer cannot be confirmed, because it is obscured by slough or eschar. If slough or eschar is removed, a Stage 3 or Stage 4 pressure injury will be revealed. Stable eschar (dry, adherent, and intact, without erythema or fluctuance) on an ischemic limb or heels should not be softened or removed. Deep tissue pressure injuries indicate intact or nonintact skin with localized areas of persistent non-blanchable deep red, maroon, purple discoloration, or epidermal separation revealing a dark wound bed or blood-filled blister. Pain and temperature changes often precede skin color changes. Discoloration may appear differently in darkly pigmented skin. This injury results from intense and/or prolonged pressure and shear forces at the bone–muscle interface. The wound may evolve rapidly to reveal the actual extent of the tissue injury or may resolve without tissue loss. If necrotic tissue, subcutaneous tissue, granulation tissue, fascia, muscles, or other underlying structures are visible, this indicates a full-thickness pressure injury [14].

Basically, as the pressure injury progresses, the category in the NPUAP classification shifts from a Stage 1 pressure injury to a Stage 4 pressure injury, and the range of the injury progresses from the epidermis to the dermis and further to the subcutaneous tissue and bone. It should be noted that the lesions are usually infected with pathogenic microorganisms, which are mainly bacteria, and the infection often contributes to the onset, progression, and recurrence of the pressure injury. Further, the infection on the pressure injury lesions often results in cellulitis, osteomyelitis, and sepsis, which, in turn, can lead to death [15,16,17,18,19,20,21,22,23]. It is thought that the majority of pressure injury infections are caused by the dermis getting exposed due to damage to the epidermis and with the exposed area getting infected with pathogenic microorganisms [15,16,17,18,19,20,21,22,23].

### 1.2. Determining the Relationship between Pressure Injury and Skin Microbiome in Bedridden Elderly People

What kind of bacteria exist in the prelesion of a pressure injury, and which bacteria contribute to the onset, progression, and recurrence of pressure injuries? Although various investigators have examined and reported on the above questions, there remains no clear answer.

There have been many reports on the existence of bacteria in the primary lesions of pressure injuries that often occur in bedridden people due to a spinal cord injury [24,25,26,27,28,29,30,31,32]. Dana et al. analyzed a number of papers, extracted via the MEDLINE database from 1996 to 2014, describing the bacteria detected on the focal of pressure injuries in people who were bedridden due to a spinal cord injury wherein *Staphylococcus aureus*, *Proteus mirabilis*, and *Pseudomonas aeruginosa* were mainly detected [32].

Moreover, Espejo et al. analyzed hospital records from 1984 to 2015 regarding the development of sepsis in hospitalized patients with bacteremia with pressure injuries [33]. According to this review study, *S. aureus*, *Proteus* spp., and *Bacteroides* spp. were detected in the blood of the majority of patients. In addition, investigating other risk factors of death from sepsis by pressure injury infections identified hospital-acquired bacteremia, polymicrobial bacteremia, and serum albumin <23 g/L as possible risk factors.

The results suggest that specific bacterial species exist in the primary lesion where pressure injury infections occur and that lethal pressure injury infections can be exacerbated by the combination of an immunocompromised situation, poor nutritional status, and unhygienic environment. However, the types of detected bacteria in the pressure injury site of bedridden people are extremely diverse; thus, the results were noted to vary significantly, depending on which method was used to detect the bacterial species [32].

In contrast, there have been a few reports examining from where these bacterial species came. Konya et al. examined the bacterial species collected from the skin at the primary pressure ulcer lesion and the healthy skin around it in bedridden elderly people and showed that there was no significant change in the bacterial species collected from both [34]. In other words, they speculated that there was a residential skin microbiome on the surface of the skin and that the infection of the bacteria in the skin microbiome to the primary pressure injury may cause severe pressure injury infections.

## 2. Examining the Skin Microbiome via Next-Generation Sequencing (NGS)

### 2.1. What Is Skin Residential Microbiome?

The skin microbiome is widely distributed on the skin surface, known to be colonized with tens to hundreds of types of bacterial species [35,36,37,38,39]. The skin microbiome, together with the stratum corneum, has been identified to function as a barrier against the infection of nonresidential pathogenic bacteria [35,36,37,38,39]. However, as reported by Espejo et al. [33], a poor nutritional status has been determined to increase the risk of death from pressure injuries, as this might change the physiological function of the skin, which, in turn, may alter the composition of the residential skin microbiome. In addition, since a poor nutritional status can also induce an immunocompromised situation, changes in the composition of the residential skin microbiome may activate some pathogenic constituent bacteria, resulting in an endogenous infection in the pressure injury lesion. However, studies have rarely investigated whether the residential skin microbiome is involved in the development of post-pressure injury infections.

### 2.2. Trends on Skin Microbiome Study, including Its Composition and Abundance Ratio

To determine the changes in the skin microbiome, it is necessary to investigate the changes in its composition and abundance ratio. In the last century, the main method of conducting a survey was to collect the microbiome existing on the skin surface via a swab method and the like, smear it on a medium, and culture it [16,20,21,22,23,24,25,26,27,36,40]. As this method cannot perform various culture conditions at the same time, it is divided into bacterial species that grow well with easily detection and bacterial species that are not [32,36,40]. As a result, it was impossible to accurately measure the bacterial species present on the skin microbiome, including their abundance ratio.

In this century, NGS analysis has made it possible to measure the bacterial species that make up the skin microbiome and their abundance ratio by a metagenomic analysis without culturing [36,41,42,43,44,45,46,47,48,49], resulting in a breakthrough of skin microbiome research. For example, several investigators, including us, have examined the composition of the residential skin microbiome in the back skin of healthy young people by NGS and found that the genera *Cutibacterium* spp., *Corynebacterium* spp., and *Staphylococcus* spp. were found to be dominant, followed by *Enhydrobacter* spp., *Flavihumibacter* spp., *Finegoldia* spp., *Methylobacterium* spp., and *Acinetobacter* spp. [50,51,52,53]. As a result of the NGS analysis, it was revealed that there are significant differences in the composition of the residential skin microbiome for each part of the body [40,54]. Furthermore, it has been clarified that *S. aureus*, which is one of the bacteria constituting the skin microbiome, is directly involved in the exacerbation of the symptoms of atopic dermatitis [55,56,57].

## 3. Skin Structure and Physiology and the Role of the Bacteria Present on the Skin Residential Microbiome on Maintaining Skin Homeostasis

The surface layer of the skin consists of severe flat epithelium and keratin, which is a fibrous protein present as a retainer of the epithelial tissue. The outermost layer is often keratinized, and lipids and fatty acids such as ceramide and cholesterol exist around the stratum corneum, with part of them exposed on the skin surface. In the epidermis, 40–50% of water is contained, and the pH of the skin surface is 4 to 5, which is weakly acidic [52,58,59,60]. As the skin is anatomically and physiologically different from the mucosal tissues in the oral cavity and intestinal tract, the composition of the residential microbiome in the skin is completely different from that in the mucosal tissues [52,61].

In the skin surface of the trunk and upper arm of healthy people, *Cutibacterium* spp. (which was formerly *Propionibacterium* spp.), *Corynebacterium* spp., and *Staphylococcus* spp. were detected, followed by various bacteria such as *Enhydrobacter* spp. and *Flavihumibacter* spp. [53,62]. For example, the major skin commensal *Cutibacterium acnes* has been known to produce fatty acids, which, in turn, keep the skin pH mildly acidic and prevent the colonization of transient bacteria [62,63]. *Staphylococcus epidermidis* is a major source of glycerol on the skin, which is responsible for skin water retention [62,64]. The depletion of *S. epidermidis* can cause skin inflammation [62,65], and several *Staphylococcus* spp. can produce antimicrobial peptides to prevent the colonization of pathogens [62,66]. On the contrary, the genera *Pseudomonas* spp. and *Streptococcus* spp., which frequently appear as the causative bacteria of skin infections, have been ranked high in skin residential microbiome, but their occupancy rate is very low [62].

Recent studies suggest that the physiological properties such as the skin water content, keratin thickness, dryness, pH, and skin organic content and topography have a significant effect on the diversity of the skin microbiome composition. Furthermore, it has been pointed out that the composition of the skin microbiome differs depending on age, health condition, and lifestyle [62,67]. It has also been reported that the composition of the skin microbiome differs significantly, depending on the site of the body of an individual [57]. On the contrary, not only bacteria but, also, fungi, viruses, and mites are present on the skin [52,54,57]. Therefore, it is considered that the skin microbiome changes depending on the physiological properties of the skin, immunity, and environmental factors surrounding the individual.

The physiological and biochemical properties of the skin are known to have a profound effect on the composition of the skin microbiome [52]. Most of the skin is cool, acidic, and dry. However, the composition of the skin microbiome changes depending on the thickness of the skin, the density of folds, hair follicles, and glands. Incisive vagina and appendages, including sweat glands, sebaceous glands, and hair follicles, are likely to be associated with a unique skin microbiome composition [68]. The eccrine gland, which is one of the sweat glands, has a function of regulating the body temperature but also has a function of excreting water and electrolytes and acidifying the skin to prevent the colonization and growth of microorganisms. On the other hand, the apocrine gland located in the axilla, papilla, and genital area also have bacterial processing and secretory-promoting functions of the eccrine glands [69,70,71,72]. Sebaceous glands, on the other hand, are connected to hair follicles to form a pilosebaceous unit and secrete lipid-rich substances. Sebum protects the skin and provides an antibacterial shield. The sebaceous glands are relatively anoxic and promote the growth of anaerobic, lipid-fed *C. acnes* and others. *C. acnes* also hydrolyzes trigluceride-containing sebum and releases free fatty acids to the skin [73,74]. Since the production of free fatty acids causes acidification of the skin, it inhibits the growth of many pathogens, such as *Staphylococcus aureus*, *Streptococcus* spp., and *Enterobacteriaceae*, and promotes the colonization and growth of coagulase-negative staphylococci and *Corynebacterium* spp. [75,76,77,78]. Therefore, in the major region of the skin, *Cutibacterium*, coagulase-negative staphylococci, and *Corynebacterium* spp. are the main components of the skin microbiome. In addition, areas with a high density of sebaceous glands, such as the face, chest, and back, tend to be inhabited by many lipophilic microorganisms such as bacteria such as *Cutibacterium* and fungi such as *Malassezia*. Skin obstructions, on the other hand, result in significant changes, including the hydration status, barrier permeability, epidermal lipids, DNA synthesis, microflora, and numerous molecular and cellular processes. The groin, axilla, and toes are prone to obstruction of the skin, are often moistened, and the pH of the skin is elevated. Therefore, many Gram-negative bacilli, coryneforms, *S. aureus*, etc. inhabit that region [79].

## 4. The Skin and the Skin Residential Microbiome among Bedridden Elderly and Their Association with the Risk of Pressure Injury

### 4.1. Skin Structure, Skin Physiology, and Skin Residential Microbiome in Elderly People

It has been reported that the physiological condition of the skin in elderly people living in nursing homes is considerably different from that of healthy people [62,67]. It was also reported that the proportion of the *Cutibacterium* spp. in the composition of the skin microbiome has been decreased in the elderly [62,80]. More interestingly, one study showed that the skin microbiome of the elderly with care support in nursing homes is very different from that of age-matched community participants [81]. The results suggest that the skin microbiome in elderly and elderly with care support can change as the physiological function of the skin also changes.

The following reports support this possibility. Some symbiotic bacteria on the skin can maintain a slightly acidic pH, and such acidic conditions can promote the growth of these symbiotic bacteria [82]. Paradoxically, pathogens can multiply when the pH of the skin is high [83]. Elderly people usually use diapers, and their skin is often in contact with urine and/or stool for a long time. Sleeping with elevated skin pH and incontinence-related skin deterioration such as dysbiosis is also a problem [84,85].

Moreover, the situation of bedridden elderly patients can be more serious than the situation of elderly patients that can support themselves. Bedridden elderly people are more prone to inconvenience in terms of performing their daily hygiene activities compared to self-supporting elderly people, with their living space and personal hygiene environment more likely to be in poor condition. In addition, they often suffer from poor nutritional status, and it is highly possible that their immunity has also weakened. Due to various factors different from those of the elderly who can support themselves, the skin microbiome composition in bedridden elderly people may be different from that of the self-supporting elderly people.

### 4.2. Characteristics of Residential Microbiome of the Dorsal Sacral Skin in Bedridden Elderly People Compared with Ambulatory Elderly People and Healthy Young People

We determined the characteristics of the skin microbiome composition in bedridden elderly people by comparing them with ambulatory elderly people and healthy young ones [62]. The measurement site was the skin of the dorsal sacral region, which is a frequent site for pressure injury. The number of bacterial genera that constitute the skin microbiome was compared via the α-diversity analysis method, wherein we found that the number of bacterial genera in the bedridden elderly group was larger than that in the other two groups [62]. The composition of the skin microbiome was almost the same between the healthy young people and ambulatory elderly people, whereas, in bedridden elderly people, *Cutibacterium* spp. (which was abundant in the two healthy groups), *Enhydrobacter* spp., *Methylobacterium* spp., and other constituent bacteria of various skin microbiomes in healthy people were noted to decrease. Instead, in bedridden elderly people, the *Corynebacterium* spp. and *Staphylococcus* spp. were found to increase, and *Enterobacteriaceae* such as *Escherichia* spp., *Shigella* spp., *Bacteroides* spp., and *Klebsiella* spp. and anaerobic bacteria that live in the intestinal tract like *Bifidobacterium* spp. were also increased [62].

Furthermore, we examined the differences of skin microbiome composition between the three groups via the heat map method. As per the results, it was determined that there was no significant difference between ambulatory elderly people and healthy young ones. Meanwhile, the skin microbiome composition in bedridden elderly was completely different from that in ambulatory elderly people and healthy young ones. The β-diversity analysis supported the data in the heat map method. Further, the β-diversity analysis suggested that this difference was influenced by the decrease in *Cutibacterium* spp. and *Enhydrobacter* spp. and the increase in *Escherichia* spp., *Shigella* spp., *Corynebacterium* spp., and *Staphylococcus* spp. in bedridden elderly people [62].

We found that there are many bacteria present in the intestinal tract in the dorsal sacral skin of bedridden elderly people. The skin surrounding the sacrum is close to the anus. Bedridden elderly people usually wear diapers, which cover the anus and sacral skin. Therefore, it is considered that the sacral skin is usually contaminated with feces and, thus, could be the reason why plenty of bacteria were detected in the intestinal tract [62]. However, the skin in this area is kept clean by nurses who make sure that the area is washed daily; it should also be noted that the area was washed again before our survey. Nevertheless, these bacteria were still detected. Furthermore, even when the same survey was conducted 2 years after the initial survey, the same skin microbiome composition as the previous survey was observed [86]. The results may suggest that the large number of intestinal bacteria present in the sacral skin of bedridden elderly people is not only due to fecal contamination but also could be attributed to a part of the contaminated intestinal bacteria fixed and inhabited on the skin. As one of the grounds for suggesting this possibility, our canonical correspondence analysis showed that a significant increase in the dorsal sacral skin pH was observed in bedridden elderly people, which makes it easier for intestinal bacteria to survive [62].

### 4.3. Association of Skin Microbiome with the Onset of Pressure Injury and Post-Pressure Injury Infections in Bedridden Elderly

It has been determined that the composition of the skin residential microbiome in bedridden elderly people is significantly different from that in ambulatory elderly people and healthy young ones. Thus, in this study, we aimed to determine whether such differences in the skin microbiome in bedridden elderly people is associated with the onset risk of a pressure injury. During the assessment of the skin residential microbiome composition in the sacral skin of bedridden elderly people, we found that three bedridden elderly people had pressure injuries in the sacral skin. Therefore, we additionally examined the skin microbiome composition at the lesions of the pressure injuries and found that the compositions of the lesions of the pressure injuries were like the compositions of the same lesions before the onset of the pressure injuries [62].

These results raised a new question: which bacteria are mostly often detected in the primary lesion during pressure injury development. Thus, it is necessary to clarify what bacteria are frequently detected before and after the onset of a pressure injury. It is well-known that pressure injuries in bedridden elderly people often recur [59]. Therefore, we investigated the frequency of the onset of recurrent pressure injuries, the physiological condition of the skin at the recurrent pressure injury lesion sites, and the bacteria that are frequently present in bedridden elderly people who have once suffered from the same condition before [59]. In total, 30 bedridden elderly who were admitted to a medical facility and who had completely recovered from a pressure injury within 1 month as regards skin physiology, composition of the skin residential microbiome, and the presence or absence of a recurrent pressure injury for 6 weeks were assessed. In this study, eight patients developed a recurrent pressure injury. The affected individuals had lower water contents in the stratum corneum than the nonaffected individuals, while the abundance ratio of *Staphylococcus* spp. was found to be significantly higher in the composition of the skin microbiome [59].

*Staphylococcus* spp. include coagulase-positive *S. aureus*, which is highly pathogenic and has a harmful effect on the skin, and coagulase-negative *S. epidermidis*, which is relatively low pathogenic and is thought to contribute to the barrier function of skin infections; in other words, *Staphylococcus* spp. include a variety of bacterial species. Therefore, it is necessary to determine which *Staphylococcus* spp. is associated with the development of recurrent pressure injuries, which we are currently investigating.

## 5. Discussion

Our previous studies [59,62] have found that the skin residential microbiome composition in bedridden elderly people is far different from that in ambulatory elderly people and healthy young ones. The major feature of the skin residential microbiome composition change among bedridden elderly people is the remarkable decreased ratio of *Cutibacterium* spp. and *Enterobacteriaceae* and increased ratio of *Escherichia* spp., *Shigella* spp., *Corynebacterium* spp., and *Staphylococcus* spp. *Cutibacterium* spp. are known to settle in lipid-rich skin areas, and these bacteria survive and proliferate by metabolizing lipids [87]. In fact, acne is a problem in puberty where lipids are most abundant in the face due to skin inflammation caused by the activation of *C. acnes*, which is a type of *Cutibacterium* spp. [62,63,87]. Since the lipid components generally present in human skin tend to decrease with aging, the decrease in *Cutibacterium* spp. can be recognized as a sign of the decrease in the lipid components of the skin. It is considered that the decrease in lipids can cause the stratum corneum to lose its moisture and weaken the barrier function of the skin. In addition, some of the lipids present in the skin have antibacterial effects against various microorganisms [87]. Therefore, decreased lipids in the skin may reduce the antibacterial effects on the skin, thus affecting the compositional changes of the skin residential microbiome.

The pH of the skin surface of bedridden elderly people tends to be higher compared to that of ambulatory elderly people and healthy young ones [62]. An increase in the pH of the epidermis of the skin reduces the water-permeable barrier function and induces the easiness of peeling of the stratum corneum. At the same time, it is considered that the increase in pH induces Th2-type inflammation, and *S. aureus* is likely to proliferate [88,89,90]. We have found that patients with recurrent pressure injuries have a high proportion of *Staphylococcus* spp. in the lesion site [59]; thus, it should be considered that the *Staphylococcus* spp., i.e., *S. aureus*, might have played a role in the onset and recurrence of pressure injuries. In addition, an increase in pH in the skin is a favorable condition for the growth of *Enterobacteriaceae*, and, as described above, *Enterobacteriaceae* bacteria might have proliferated due to the skin being contaminated with feces. However, it remains completely unclear how they are involved in the development of pressure injuries, which should be a subject for further study.

Our study shows that the composition of the microbiome on the sacral skin, which is an area of pressure in bedridden elderly people, is very different from that of healthy young and ambulant elderly people [62]. In the sacral skin microbiome in bedridden elderly people, the abundance ratio of *Cutibacterium* spp. is catastrophically reduced, and the abundance ratios of *Staphylococcus* spp. and *Enterobacteriaceae* are increased. Further, the composition of the microbiome of the part where the pressure injury developed retained its characteristics. *Cutibacterium* spp. is abundant in the skin rich in sebaceous glands. The *Cutibacterium* spp. hydrolyzes sebum-bearing trigluceride to produce free fatty acids and acidify the skin. Therefore, it is considered that the skin rich in *Cutibacterium* spp. has a strong barrier function against the invasion of external bacteria by the abundant sebum, and the acidification of the skin suppresses the colonization and growth of many pathogenic bacteria. However, it is possible that the skin of bedridden elderly people, whose *Cutibacterium* spp. are catastrophically reduced, loses these two functions. It has been clarified that the pH of the skin is significantly higher in bedridden elderly people than in healthy young people and ambulatory elderly people. Therefore, the abundance ratio of pathogenic *Staphylococcus* and *Enterobacteriaceae* in the sacral skin of bedridden elderly people is increasing. Epithelial cells are often damaged in the skin of pressure injury-prone areas due to a lack of blood flow due to long-term pressure. Therefore, it is considered that the loss of two functions in this part facilitates the infection of a damaged epithelium with various pathogenic bacteria and facilitates the development of pressure ulcers. We also found that the composition of the skin microbiome on the contralateral region of bedridden elderly people who developed pressure injuries was not different from that of the non-pressure injury region. Similar results could be seen in the study by deWert et al. [91]. However, in our study, we limited the measurement area to the sacral region, because we first focused on the relationship between the pressure injury and skin parameters, such as the skin microbiome and skin physiological functions. As the compositions of skin microbiome greatly differ depending on the area [52], and the same is true for older patients [81], other parts of the body should be of interest. Changes in the skin microbiome in other regions need to be investigated in the future.

In bedridden elderly people, the composition of the skin microbiome in the sacral region is significantly different from that in ambulatory elderly people and healthy young people, and the same result in the back skin region is also observed [62]. These results suggest that the constitutional change of the skin microbiome in bedridden elderly people may be observed in the skin other than the pressure region. However, it is unknown whether similar changes are observed in parts other than the back, because we have not considered the issue yet.

It has already been reported that the composition of the intestinal microbiome is significantly altered by the administration of various antibiotics [92,93,94]. However, few reports have evaluated the changes in the composition of the skin microbiome due to antibiotic administration [52]. Therefore, it is impossible to mention the role of antibiotics in the skin flora from the pressure region. In our study presented here [59,62], none of the pressure injury patients and bedridden elderly people surveyed received antibiotics. Therefore, we do not really understand this either. In the future, it will be necessary to investigate the changes in the composition of the bacterial flora present at the pressure ulcer site and the presence or absence of healing due to local or systemic antibiotic administration.

Skin residential microorganisms include not only bacteria but also fungi and mites [95,96]. *Malassezia* spp. is a well-known fungus that resides on the skin, and the *Demodex* spp. is a well-known mite that resides on the skin. The *Demodex* spp. is found throughout the skin, centered on the face [95,96,97]. Although there is no direct scientific evidence, it is believed to be associated with the development of Rosacea [95,96,97]. It has been reported that allergies, diabetes mellitus, obesity, and old age are factors that increase the abundance ratio of *Demodex folliculorum* in the eyelashes [98]. Another important feature of *Demodex* is the endosymptom of bacteria such as *Bacillus oleronius*, *B. simplex*, *B. pumilus*, *B. cereus*, and *Corynebacterium kroppenstedtii* subsp. *demodicis* [99,100,101,102]. Therefore, the presence of *Demodex* suggests the possibility of spreading the bacteria contained on the skin. Few reports have examined the role of *Demodex* in pressure injuries. It is thought that pressure injuries rarely develop in the facial area. However, *Demodex* tends to increase with aging, and it cannot be denied that it may also inhabit pressure injury-prone sites. There are many microorganisms inside *Demodex*, and it is possible that these bacteria will be released and settle on the skin. Whether or not the skin infections caused by pressure injuries worsen in that situation should be examined in the future.

We confirmed that the composition of the skin microbiome present at pressure injuries is the same as the composition that was present before pressure injuries are formed [62]. Therefore, it is considered important to regularly wash the skin to control the number of bacteria constituting the microbiome in order to prevent infection of the pressure injury. In addition, there are reports suggesting that the bedridden environment (i.e., bed sheets, pillow covers, etc.) used by bedridden elderly people may be important for the infection of pressure injuries. The report shows that the changes over time in the microbiome present in sheets used by bedridden elderly people with pressure injuries are greater than those used by bedridden elderly people without pressure injuries [103], suggesting that this change in the microbiome can cause infections from pressure injuries. In addition, there are reports that people who wash their face less often and use shared face towels instead of privately owned ones after washing their faces have a high prevalence of *Denodex folliculorum* in their facial skin [97]. The result predicts that frequent cleaning of the skin keeps it clean and that towels and sheets that come into contact with the skin should not be shared, and frequent washing is performed to improve hygiene, which can lead to pressure injuries and wound infections.

## 6. Conclusions

Bedridden elderly people were identified to be at higher risk of developing pressure injuries due to weakened immunities and skin functions caused by poor nutrition and decreased physiological functions. However, as per the results of our study, it is possible that the composition of the skin microbiome in bedridden elderly people, which is different from that in ambulatory elderly people and healthy young ones, and the increase in the abundance ratio of pathogenic bacteria such as *Staphylococcus* spp. may be considered the main factors contributing to a pressure injury and post-pressure injury infections. To determine the mechanisms of pressure injuries among bedridden elderly people, it is necessary to identify the characteristics of the changes in the composition of the skin microbiome in addition to the skin physiology and immunity, as well as to determine the interrelationship of these three factors in the onset of a pressure injury and post-pressure injury infections.

## Data Availability

Not applicable.

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
