# Peer review of "Association of Skin Microbiome with the Onset and Recurrence of Pressure Injury in Bedridden Elderly People"

_microorganisms, 2021, doi:10.3390/microorganisms9081603_

Round 1

Reviewer 1 Report

This manuscript by  Shigefumi Okamoto et al. describes very interesting data on the association of skin microbiome with the onset /recurrence of pressure injury in bedridden people.

This manuscript suffer from few limitation. Revision is needed.

Line 38 : website have to be refereced as other bibliographic reference.

"i.e." and "et al." have to be italicized.

Part 3. lack some information about the numeric importance of the skin microbiome and difference betwe dry and wet skin part of the anatomy.

Author Response

Response to Reviewer 1 Comment

This manuscript by Shigefumi Okamoto et al. describes very interesting data on the association of skin microbiome with the onset /recurrence of pressure injury in bedridden people.

Response

We thank the reviewer for your appreciation for the content of our manuscript.

This manuscript suffer from few limitation. Revision is needed.

Line 38 : website have to be refereced as other bibliographic reference.

Response

We added the website to the reference list (Ref. 1 in the revised manuscript).

"i.e." and "et al." have to be italicized.

Response

We thank the reviewer for careful review. We corrected “i.e.” and “et al.” to italicize.

Part 3. lack some information about the numeric importance of the skin microbiome and difference between dry and wet skin part of the anatomy.

Response for Part 3

We thank the reviewer for indicating important point. According to the reviewer’s suggestion, we added the following paragraph in the revised manuscript (Page 5, Lines 211-238 in the revised manuscript).

The physiological and biochemical properties of the skin are known to have a profound effect on the composition of the skin bacterial flora [52]. Most of the skin is cool, acidic and dry. However, the composition of the skin microbiome changes depending on the thickness of the skin, the density of folds, hair follicles, and glands. Incisive vagina and appendages, including sweat glands, sebaceous glands, and hair follicles, are likely to be associated with a unique skin microbiome composition [68]. The eccrine gland, which is one of the sweat glands, has a function of regulating body temperature, but also has a function of excreting water and electrolytes and acidifying the skin to prevent colonization and growth of microorganisms. On the other hand, the apocrine gland located in the axilla, papilla, and genital area also have bacterial processing and secretory promoting functions of the eccrine glands [69-72]. Sebaceous glands, on the other hand, are connected to hair follicles to form pilosebaceous unit and secrete lipid-rich substances. Sebum protects the skin and provides an antibacterial shield. The sebaceous glands are relatively anoxic and promote the growth of anaerobic, lipid-fed Cutibacterium acnes and others. C. acnes also hydrolyzes trigluceride-containing sebum and releases free fatty acids to the skin [73, 74]. Since the production of free fatty acids causes acidification of the skin, it inhibits the growth of many pathogens such as Staphylococcus aureus, Streptococcus spp., and Enterobacteriaceae, and promotes the colonization and growth of coagulase-negative staphylococci and Corynebacterium spp. [75-78]. Therefore, in major region of the skin, Cutibacterium, coagulase-negative staphylococci, and Corynebacterium spp. are the main components of the skin microbiome. In addition, areas with high density of sebaceous glands, such as the face, chest, and back, tend to be inhabited by many lipophilic microorganisms such as bacteria such as Cutibacterium and fungi such as Malassezia. Skin obstruction, on the other hand, results in significant changes including hydration status, barrier permeability, epidermal lipids, DNA synthesis, microflora, and numerous molecular and cellular processes. The groin, axilla, and toes are prone to obstruction of the skin, are often moistened, and the pH of the skin is elevated. Therefore, many gram-negative bacilli, coryneforms, Staphylococcus aureus, etc. inhabit that region [79].

Reviewer 2 Report

1.The authors should discuss if the skin microbioma changes on pressure areas can influence and how the skin microbioma of contralateral or other skin areas at distance.

2.The authors should mention the role of sistemic or topical antibiotics used for  those peoples on skin microbioma from pressure areas.

3.Did authors find influences arround the pressure areas?

4.One of the residents of the skin is Demodex Folliculorum.The authors are invited to discuss data related to Demodex from those areas,facial and nonfacial areas where Demodex was also described

5.Some of the patients who are in bedridden state have comorbidiries like Rosacea or Rosacea like Demodicosis.Demodex Folliculorum is also a host for some previously described endosymbionts like Bacillus Oleronius,Simplex,Pumilus,Cereus.Please comment and reference those situations.

5.If possible ,please add information about facial skin microbioma changes,on facial or latero facial pressure areas induced by pillows or other covering sheets,or towels for facial or other areas hygiene.

Author Response

Response to Reviewer 2 Comment

1.The authors should discuss if the skin microbiota changes on pressure areas can influence and how the skin microbiota of contralateral or other skin areas at distance.

Response for 1

Thank you for carefully reviewing our manuscript. We consider the following hypothesis from our research results on how changes in the microbiome on the sacral skin, which is the pressure area of bedridden elderly people, affect the development of pressure injury.

Our study shows that the composition of the microbiome on the sacral skin, which is the area of pressure in bedridden elderly, is very different from that of healthy young and ambulant elderly [62]. In the sacral skin microbiome in the bedridden elderly, the abundance ratio of Cutibacterium spp. is catastrophically reduced, and the abundance ratio of Staphylococcus spp. and Enterobacteriaceae is increased. In further, the composition of the microbiome of the part where the pressure injury developed retained its characteristics. Cutibacterium spp. is abundant in the skin rich in sebaceous glands. The Cutibacterium spp. hydrolyzes sebum-bearing trigluceride to produce free fatty acids and acidify the skin. Therefore, it is considered that the skin rich in Cutibacterium spp. has a strong barrier function against the invasion of external bacteria by the abundant sebum, and the acidification of the skin suppresses the colonization and growth of many pathogenic bacteria. However, it is possible that the skin of bedridden elderly people, whose Cutibacterium spp. are catastrophically reduced, loses these two functions. It has been clarified that the pH of the skin is significantly higher in bedridden elderly people than in healthy young people and ambulatory elderly people. Therefore, the abundance ratio of pathogenic Staphylococcus and Enterobacteriaceae in the sacral skin of bedridden elderly people is increasing. Epithelial cells are often damaged in the skin of pressure injury-prone areas due to lack of blood flow due to long-term pressure. Therefore, it is considered that the loss of two functions in this part facilitates the infection of damaged epithelium with various pathogenic bacteria and facilitates the development of pressure ulcers. We also found that the composition of the skin microbiome on the contralateral region of the bedridden elderly who developed pressure injury was not different from that of the non-pressure injury region. Similar results can be seen in the study by deWert et al. [91]. However, in our study, we have limited the measurement area to the sacral region because we first focused on the relationship between PI and skin parameters such as skin microbiome and skin physiological functions. As the compositions of skin microbiome greatly differ depending on the area [52] and the same is true for older patients [92], other parts of the body should be of interest. Changes in the skin microbiome in other region need to be investigated in the future.

The above description has been added to the revised manuscript (Page 8, Lines 370-398 in the revised manuscript).

2.The authors should mention the role of systemic or topical antibiotics used for those peoples on skin microbiota from pressure areas.

Response for 2

It has already been reported that the composition of the intestinal microbiome is significantly altered by the administration of various antibiotics [93-95]. However, few reports have evaluated changes in the composition of the skin bacterial flora due to antibiotic administration [52]. Therefore, it is impossible to mention the role of antibiotics in the skin flora from the pressure region. In our study presented here [59, 62], none of the pressure injury patients and bedridden elderly people surveyed received antibiotics. Therefore, we don't really understand this either. In the future, it will be necessary to investigate changes in the composition of the bacterial flora present at the pressure ulcer site and the presence or absence of healing due to local or systemic antibiotic administration.

The above has been added to the revised manuscript (Page 8, Line 406-Page 9, Line 414 in the revised manuscript).

3.Did authors find influences arround the pressure areas?

Response for 3

In bedridden elderly people, the composition of the skin microbiome in the sacral region is significantly different from that in ambulatory elderly people and healthy young people, and it is also observed same result in the back skin region [62]. The result suggests that the constitutional change of the skin microbiome in the bedridden elderly may be observed in the skin other than the pressure region. However, it is unknown whether similar changes are observed in parts other than the back, because we have not considered the issue yet.

The above has been added to the revised manuscript (Page 8, Lines 399-405).

4.One of the residents of the skin is Demodex Folliculorum. The authors are invited to discuss data related to Demodex from those areas, facial and nonfacial areas where Demodex was also described.

5.Some of the patients who are in bedridden state have comorbidiries like Rosacea or Rosacea like Demodicosis. Demodex Folliculorum is also a host for some previously described endosymbionts like Bacillus Oleronius,Simplex,Pumilus,Cereus.Please comment and reference those situations.

Response for 4 and 5

Thank you for pointing out. There is no report on the relationship between Demodex and pressure ulcers, but I think it is a good question to be investigated in the future. Therefore, I added the following comments to the discussion (Page 9, Lines 415-430).

Skin residential microorganisms include not only bacteria but also fungi and mites [96, 97]. Malassezia spp. is a well-known fungus that resides on the skin, and the Demodex spp. is a well-known mite that resides on the skin. The Demodex spp. is found throughout the skin, centered on the face [96-98]. Although there is no direct scientific evidence, it is believed to be associated with the development of Rosacea [96-98]. It has been reported that allergies, diabetes mellitus, obesity, and old age are factors that increase the abundance ratio of Demodex folliculorum in eyelashes [99]. Another important feature of Demodex is the endosymptom of bacteria such as Bacillus oleronius, B. simplex, B. pumilus, B. cereus, and Corynebacterium kroppenstedtii subsp. demodicis [100-103]. Therefore, the presence of Demodex suggests the possibility of spreading the bacteria contained on the skin. Few reports have examined the role of Demodex in pressure injury. It is thought that pressure injury rarely develops in the facial area. However, Demodex tends to increase with aging, and it cannot be denied that it may also inhabit pressure injury-prone sites. There are many microorganisms inside Demodex, and it is possible that these bacteria will be released and settle on the skin. Whether or not the skin infection caused by pressure injury worsens in that situation should be examined in the future.

6.If possible, please add information about facial skin microbiota changes, on facial or latero facial pressure areas induced by pillows or other covering sheets, or towels for facial or other areas hygiene.

Response for 6

Thank you for your advice. This may be an inadequate answer due to the very small amount of references available to answer this question, but we have summarized it below and added it to my discussion (Page 9, Lines 431-446).

We have confirmed that the composition of the skin microbiome present at the pressure injury is the same as the composition that was present before the pressure injury was formed [62]. Therefore, it is considered important to regularly wash the skin to control the number of bacteria constituting the microbiome in order to prevent infection of the pressure injury. In addition, there are reports suggesting that the bedridden environment (i.e. bed sheets, pillow cover etc.) used by bedridden elderly people may be important for infection of pressure injury. The report shows that the changes over time in the microbiome present in sheets used by bedridden elderly people with pressure injury are greater than those used by bedridden elderly people without pressure injury [104], suggesting that this change in microbiome can cause infection of pressure injury. In addition, there are reports that people who wash their face less often and use shared face towels instead of privately owned ones after washing their faces have a high prevalence of Denodex folliculorum in their facial skin [98]. The result predicts that frequent cleaning of the skin keeps it clean, and that towels and sheets that come into contact with the skin are not shared, and frequent washing is performed to improve hygiene, which can lead to pressure injury and wound infections.